# Sparse Hyperbolic Convolutional Networks with Enhanced Object Localization via GradCAM Analysis

Vijayavallabh Jayamanikandan

be23b041@smail.iitm.ac.in

Jithamanyu Settur

sjithamanyu@gmail.com

Lokesh Rajulapati

lokesh.rajulapati@gmail.com

Raghunathan Rengaswamy

raghur@dsai.iitm.ac.in

*Indian Institute of Technology Madras, India*

## Abstract

*Hyperbolic spaces model hierarchical structures within data. Studies have demonstrated that spatial representations in the hippocampus are structured within hyperbolic spaces to optimize efficiency[26]. We explore the use of hyperbolic convolutional networks with sparsity constraints (L1 and Top-k) and analyze the significance of features in the images for classification tasks using GradCAM. We show that applying sparsity constraints to hyperbolic convolutional networks yields performance comparable to established benchmarks and results in greater interpretability. This work develops sparse hyperbolic representations, enhancing interpretability in AI systems. Link to our code*

## 1. Introduction

Deep convolutional neural networks have revolutionised computer vision by learning hierarchical feature representations that capture complex visual patterns. However, traditional CNNs operate exclusively in Euclidean space, fundamentally limiting their ability to model the inherent hierarchical structures present in visual data [7, 20]. Real-world images exhibit rich hierarchical organizations, from fine-grained textures to object parts, from parts to complete objects that would benefit from geometric spaces designed to naturally accommodate such tree-like structures. Recent neuroscience experiments reveal that spatial representations in CA1 hippocampal neurons in rats are organized within hyperbolic spaces, allowing efficient coding that expands dynamically over time[26].

Hyperbolic geometry, characterized by constant negative curvature, offers a compelling alternative to Euclidean representations. Unlike flat Euclidean space, hyperbolic space exhibits exponential volume growth, making it particularly well suited for embedding hierarchical data with minimal distortion. Recent advances in hyperbolic neural networks have demonstrated significant improvements in tasks that involve hierarchical data, such as knowledge graphs and social networks [4]. However, the application of hyperbolic geometry to standard computer vision tasks remains largely underexplored, with most existing work focusing on specialized domains or requiring architectural constraints that limit practical applicability.

A critical challenge in modern deep learning is interpretability. Experimental evidence in neuroscience suggests that the constraints on the energy budget tries to drive the brain towards energy-efficient neural codes and wiring patterns, resulting in sparse codes[1]. Sparsity mechanisms offer a promising solution by selectively retaining only the most informative features while eliminating redundant parameters [13]. Two primary approaches have emerged: L1 regularization, which naturally induces sparsity through geometric properties of the L1 norm, and Top-K selection, which provides direct control over sparsity levels by retaining only the most significant activations [9]. Although these techniques have been extensively studied in Euclidean neural networks, their application to hyperbolic architectures remains unexplored.

Understanding and interpreting the decision-making processes of deep neural networks has become increasingly important as these models are deployed in critical applications. Gradient-weighted Class Activation Mapping (GradCAM) has emerged as a powerful tool to provide visual explanations by highlighting regions in input images that contribute the most significantly to model predictions [22]. However, existing interpretability methods are designed exclusively for Euclidean networks and do not account for the unique geometric properties and constraints of hyperbolic space. This limitation prevents us from understanding how hyper-

bolic networks make decisions and whether their purported advantages in hierarchical modeling translate to improved attention mechanisms in computer vision tasks.

**Our Contributions.** In this work, we address these limitations by introducing the first comprehensive framework for sparse hyperbolic convolutional neural networks with enhanced interpretability. Our key contributions are as follows.

1. **Sparse Hyperbolic CNNs:** We present novel implementations of L1 regularization and Top-K sparsity mechanisms specifically designed for hyperbolic convolutional neural networks operating in the Lorentz model, which act on the activations making the activations sparser. Our approach maintains the geometric constraints of hyperbolic space while achieving sparsification.

2. **Hyperbolic GradCAM:** We extend gradient-weighted class activation mapping to work with hyperbolic neural networks by decomposing gradients and activations into temporal and spatial components that respect the underlying Lorentzian geometry. This enables visual interpretation of sparse hyperbolic network decisions for the first time.

3. **Comprehensive Comparative Analysis:** We provide the first systematic comparison between sparse Euclidean ResNet architectures and their hyperbolic counterparts using both traditional performance metrics and visual explanation analysis. Our experiments on CIFAR-10 and CIFAR-100 demonstrate that sparse hyperbolic networks consistently achieve superior object localization compared to their Euclidean equivalents.

Our experimental results on CIFAR-10 and CIFAR-100 demonstrate that hyperbolic CNNs with both L1 and Top-K sparsity constraints outperform their Euclidean counterparts in terms of object localization quality, as evidenced by GradCAM visualizations that show more precise and semantically meaningful attention patterns.

## 2. Background

This section outlines the key theoretical foundations underlying our work: hyperbolic geometry and its relevance for deep learning, hyperbolic convolutional neural networks (HCNNs), sparsity mechanisms in neural representations, and gradient-based visual explanation methods. Together, these components motivate and enable the design of interpretable and efficient hyperbolic models for visual recognition tasks.

### 2.1. Hyperbolic Geometry for Deep Learning

Hyperbolic geometry is a non-Euclidean space of constant negative curvature, offering a natural inductive bias for representing hierarchical and tree-like structures often found in linguistic and visual data [6, 20].

**Lorentz Model.** We adopt the Lorentz (or hyperboloid) model for its numerical stability in optimization and compatibility with Riemannian geometry toolkits [15]. The $d$-dimensional hyperbolic space $\mathbb{H}^d$ is realized as:

$$\mathbb{H}^d = \left\{ x \in \mathbb{R}^{d+1} : \langle x, x \rangle_L = -1, \ x_0 > 0 \right\} \quad (1)$$

where the Lorentzian inner product is defined as:

$$\langle x, y \rangle_L = -x_0 y_0 + \sum_{i=1}^{d} x_i y_i \quad (2)$$

Key operations include the exponential map $\exp_x^L : T_x\mathbb{H}^d \to \mathbb{H}^d$ and logarithmic map $\log_x^L : \mathbb{H}^d \to T_x\mathbb{H}^d$, which bridge the manifold and its tangent space:

$$\exp_x^L(v) = \cosh(\|v\|_L)x + \sinh(\|v\|_L)\frac{v}{\|v\|_L} \quad (3)$$

$$\log_x^L(y) = d_L(x, y) \cdot \frac{y + \langle x, y \rangle_L x}{\|y + \langle x, y \rangle_L x\|_L} \quad (4)$$

where $d_L(x, y) = \operatorname{arccosh}(-\langle x, y \rangle_L)$ is the Lorentzian geodesic distance.

### 2.2. Hyperbolic Convolutional Neural Networks

While standard convolutional neural networks (CNNs) operate in Euclidean space, their representational capacity is limited when modeling inherently hierarchical visual structures. Hyperbolic CNNs extend standard convolutions to curved spaces by operating in tangent spaces via Riemannian mappings [4, 23].

A typical hyperbolic convolution consists of three stages:

$$\tilde{f}(y_i) = \log_x^L(f(y_i)) \text{ (Map features to tangent space)} \quad (5)$$

$$\tilde{g}(x) = \sum_i k_i \tilde{f}(y_i) \text{ (Euclidean-like convolution)} \quad (6)$$

$$g(x) = \exp_x^L(\tilde{g}(x)) \text{ (Map back to manifold)} \quad (7)$$

For computational efficiency, they adopt a linearized kernel formulation by expressing 2D convolution as:

$$\text{LConv2d}(x) = \text{LFC}(\text{Unfold}(x)) \quad (8)$$

where Unfold extracts spatial patches and LFC denotes Lorentz fully connected operations. Temporal components are handled via a rescaling procedure:

$$x_{\text{time}}^{\text{rescaled}} = \sqrt{\sum x_{\text{time}}^2 - (k_{\text{len}} - 1) \cdot \kappa} \quad (9)$$

where $\kappa$ is the curvature of the hyperbolic space and $k_{\text{len}}$ is the number of time like dimensions.

To maintain numerical stability and preserve the manifold geometry, batch normalization is performed in the tangent space. Given input $x$, we compute the Fréchet mean $\mu$ and perform:

$$x_T = \log_\mu^L(x) \tag{10}$$

$$\hat{x}_T = \gamma \frac{x_T - \mu_T}{\sqrt{\sigma_T^2 + \epsilon}} + \beta \tag{11}$$

$$\hat{x} = \exp_\mu^L(\hat{x}_T) \tag{12}$$

Here, $\mu_T$ and $\sigma_T^2$ are the mean and variance in the tangent space, and $\gamma, \beta$ are learnable affine parameters.

Finally, classification is performed using hyperbolic hyperplanes defined in Lorentz space. For each class $c$, the hyperplane is parameterized by a scalar offset $a_c \in \mathbb{R}$ (which controls the hyperplane's distance from the origin, analogous to a bias term) and a direction vector $z_c \in \mathbb{R}^d$ (which determines the orientation of the hyperplane in Euclidean coordinates). From these parameters, the Lorentzian weight vector and the logit is computed as:

$$w_{t,c} = \sinh(\sqrt{\kappa^{-1}}\, a_c)\, \|z_c\|, \tag{13}$$

$$w_{s,c} = \cosh(\sqrt{\kappa^{-1}}\, a_c)\, z_c, \tag{14}$$

$$\text{logit}_c = -\langle w_c, x \rangle_L. \tag{15}$$

## 2.3. Gradient-weighted Class Activation Mapping (GradCAM)

GradCAM [22] is a widely used technique for visual model explanation. It highlights input regions that most influence a model's prediction for a specific class $c$, based on gradient information. Given a feature map $A^k$ and the gradient of the output score $y^c$ with respect to $A^k$, the class-specific importance weight is computed as:

$$\alpha_k^c = \frac{1}{Z} \sum_{i,j} \frac{\partial y^c}{\partial A_{ij}^k} \tag{16}$$

The GradCAM localization map is then given by:

$$L_{\text{GradCAM}}^c = \text{ReLU}\left( \sum_k \alpha_k^c A^k \right) \tag{17}$$

In our work, we generalize GradCAM to hyperbolic settings by accounting for curvature and the temporal-spatial decomposition inherent in Lorentzian embeddings. This allows us to evaluate the interpretability of sparse hyperbolic networks through visual explanations that respect the geometry of the representation space.

## 3. Related Work

Our work lies at the intersection of hyperbolic geometry in vision, sparse neural networks, and interpretability techniques. We provide a comprehensive review of the most relevant contributions across these domains.

### 3.1. Hyperbolic Geometry in Computer Vision

Hyperbolic geometry has gained significant traction in computer vision due to its exponential volume growth and natural capacity to model hierarchical structures [16, 20]. The field has seen substantial progress in recent years, with comprehensive surveys highlighting the potential of hyperbolic embeddings for various vision tasks [17]. Chami et al. [4] demonstrated that hyperbolic graph neural networks preserve hierarchical information more effectively than their Euclidean counterparts, establishing fundamental theoretical foundations.

Building upon these insights, Bdeir et al. [2] introduced HCNN, the first fully hyperbolic convolutional neural network capable of performing hyperbolic batch normalization and classification directly in the Lorentz model. This work addressed limitations of earlier approaches, such as Van Spengler et al. [24], which used expansive convolutions in the Poincaré disk but relied on architectural constraints limiting practical applicability.

The theoretical understanding has also advanced significantly. Li et al. [10] conducted a comprehensive analysis of why hyperbolic neural networks are effective, proposing benchmarks for evaluating hierarchical representation capability and demonstrating that current methods cannot achieve optimal embeddings. This work reveals that hierarchical representation capability is significantly affected by optimization objectives and hierarchical structures.

Specialized applications have emerged across diverse domains in 2024. Zhang et al. [27] applied fully hyperbolic neural networks to study ageing trajectories in brain networks using magnetoencephalography data from 587 individuals, revealing that hyperbolic features outperform traditional graph-theoretic measures in capturing age-related information.

### 3.2. Sparsity in Neural Networks

Sparsity in neural networks can be imposed not only on parameters, but also on *activations*, which has proven especially important for representation learning. Activation sparsity encourages only a small fraction of neurons to be active for any given input, leading to more efficient, robust, and interpretable models. A classic example is the sparse autoencoder [19], where a sparsity penalty drives hidden units toward low average activation, forcing the network to learn more discriminative features. Extensions such as $k$-sparse autoencoders [14] instead enforce sparsity by keeping only the top-$k$ activations while suppressing the rest.

Activation sparsity has been shown to improve generalization under limited data, reduce redundancy, and enhance feature selectivity [8, 25]. Practical implementations often use $L_1$ penalties on hidden activations or top-$K$ masking functions to enforce controlled sparsity. Despite these advances in Euclidean settings, activation sparsity mecha-

nisms remain largely unexplored in hyperbolic neural networks. In this work, we introduce $L_1$ and Top-$K$ activation sparsity specifically adapted for hyperbolic CNNs in the Lorentz model, ensuring that sparsity is imposed while preserving the underlying geometric constraints.

### 3.3. Visual Explanation Techniques

GradCAM [22] and its variants [5] have become fundamental tools for visual model explanation, with numerous enhancements proposed in recent literature. The field has expanded significantly with hybrid approaches and domain-specific applications.

The integration of GradCAM with modern architectures has also progressed. Navarro-Ramirez et al. [18] evaluated different CAM variants across various CNN architectures, showing that ConvNext models produce less variable CAM maps. Layer-wise interpretability studies have gained attention, with Liu et al. [12] exploring the impact of different layer types on model interpretability using Grad-CAM for facial expression recognition.

While these hybrid approaches have explored combining GradCAM with techniques like LRP [3], no existing work extends GradCAM to hyperbolic networks. We propose Hyperbolic GradCAM to fill this gap, enabling manifold-aware interpretation of sparse Lorentz-based models for the first time.

## 4. Methods

**Building on Prior Work.** Leveraging the Lorentz model's stability and the effectiveness of fully hyperbolic convolutional architectures [2, 4, 7, 15], we adopt this foundation to construct our hyperbolic networks. Our contributions extend this line of work by introducing sparsity-driven mechanisms for disentanglement and interpretability in hyperbolic space, along with a novel adaptation of GradCAM tailored to the Lorentzian geometry.

### 4.1. Sparsity-Induced Interpretable Representations in Hyperbolic Networks

To promote interpretability in hyperbolic space, we introduce sparsity into our model via two mechanisms: L1 regularization and Top-K activation masking. Sparse representations have been shown to improve interpretability and generalization [8, 13, 21], and we adapt these principles to the Lorentzian manifold.

**L1 Regularization in Hyperbolic Space.** Given hyperbolic activations $h \in \mathcal{H}^d$, we apply L1 regularization directly in hyperbolic space. The rationale for direct hyperbolic L1 sparsity is threefold: (1) **Geometric consistency**: Operating on the manifold respects exponential volume growth properties, avoiding tangent space projection distortions. (2) **Hierarchical preservation**: Maintains

structural relationships encoded through geodesic distances during feature selection.

We decompose hyperbolic activations into temporal and spatial components, computing the L1 penalty on spatial components:

$$h_{\text{time}} = h[:, :, 0], \quad h_{\text{space}} = h[:, :, 1 :] \quad (18)$$

$$\mathcal{L}_{\text{sparse}} = \mathcal{L}_{\text{task}} + \lambda \cdot \text{mean}(|h_{\text{space}}|) \quad (19)$$

This formulation exploits the Lorentz model geometry. Points satisfy $\langle h, h \rangle_L = -h_0^2 + \|h_{\text{space}}\|^2 = -1$ with $h_0 > 0$. Since $h_0 = \sqrt{1 + \|h_{\text{space}}\|^2}$ is determined by spatial components, these represent the true degrees of freedom. Regularizing only spatial components controls intrinsic complexity without constraining the temporal component required for the hyperboloid constraint.

**Top-K Activation Masking.** To impose structured sparsity, we experiment with forwarding only the top-k activations. However, unlike conventional Euclidean approaches that select based on magnitude, we implement Top-K sparsity directly in hyperbolic space to respect the underlying geometry.

The equivalent of performing Top-K selection in hyperbolic space is to select activations based on their hyperbolic geodesic distance from the origin. For a point $x \in \mathcal{H}^d$ in hyperbolic space, the geodesic distance from the origin is given by:

$$d_{\text{hyp}}(x, 0) = \text{arccosh}(-\langle x, 0 \rangle_L) \quad (20)$$

We select $\rho\%$ of activations from the total number of activations based on this hyperbolic distance metric:

$$k = \lfloor \rho \cdot n \rfloor \quad (21)$$

$$\text{TopK}_\rho^{\text{hyp}}(h)_i = \begin{cases} h_i & \text{if } d_{\text{hyp}}(h_i, 0) \text{ in top-k} \\ 0 & \text{otherwise} \end{cases} \quad (22)$$

To validate this approach, we performed an empirical analysis comparing conventional Top-K selection (based on activation magnitude) with our hyperbolic geodesic distance-based selection. Using random point sampling across the hyperbolic manifold, we computed the correlation between these two selection criteria. Our analysis reveals a strong positive correlation $> 0.98$ between Top-K selection based on activation magnitude and Top-K selection based on hyperbolic geodesic distance from the origin, justifying our approach as a geometrically principled equivalent.

The final sparse hyperbolic activations are computed as:

$$h_{\text{topk}} = \text{TopK}_\rho^{\text{hyp}}(h) \quad (23)$$

Gradients are propagated through the discrete Top-K operation via straight-through estimation:

$$\frac{\partial \mathcal{L}}{\partial h} = \frac{\partial \mathcal{L}}{\partial \text{TopK}^{\text{hyp}}(h)} \cdot I_{\text{selected}} \quad (24)$$

where $I_{\text{selected}} \in \{0,1\}^d$ is an indicator mask with ones at the indices chosen by Top-K and zeros elsewhere. This hyperbolic-aware Top-K selection ensures that sparsity constraints respect the exponential volume growth properties of hyperbolic space, leading to more geometrically consistent sparse representations compared to naive magnitude-based selection in the tangent space. We have restricted the top-k sparsity currently to the later layer of the model. We plan to build on this in coming works.

### 4.2. Hyperbolic GradCAM for Visual Explanation

To evaluate the interpretability benefits of sparsity in hyperbolic neural networks, particularly for vision tasks, we extend the well-established GradCAM technique [22] to the Lorentzian setting. Our proposed Hyperbolic GradCAM respects the manifold structure and disentangles spatial and temporal contributions to enable geometry-aware visualizations. This extension is crucial because standard GradCAM is designed for Euclidean spaces, where features are flat and uniform. In contrast, hyperbolic spaces have constant negative curvature, as you move away from a centre, much like how a tree branches out rapidly. This makes them ideal for capturing hierarchical structures in data, but it requires adapting explanation methods to account for this curved geometry.

Intuitively, these methods refine how we "highlight" important parts of an image; our hyperbolic version builds on this by incorporating the space's curvature to ensure the highlights respect the hierarchical nature of the features.

#### 4.2.1. Temporal-Spatial Decomposition

Given hyperbolic activations $A \in \mathbb{R}^{H \times W \times C}$ and gradients $G \in \mathbb{R}^{H \times W \times C}$ in Lorentz space (with $C \geq 2$), where $H$, $W$, $C$ refer to the height, width, and number of channels of the filter outputs, we decompose each into temporal and spatial components:

$$A_{\text{time}} = A[:,:,0], \quad A_{\text{space}} = A[:,:,1:] \quad (25)$$
$$G_{\text{time}} = G[:,:,0], \quad G_{\text{space}} = G[:,:,1:] \quad (26)$$

In the Lorentz model of hyperbolic space, the "temporal" component (index 0) acts like a time dimension in special relativity, anchoring the position on the hyperboloid, while the "spatial" components (indices 1+) capture the actual features. Separating them allows us to handle the unique geometry: the temporal part ensures points stay on the manifold, preventing distortions, while the spatial part focuses on the hierarchical embeddings.

#### 4.2.2. Curvature-Aware Importance Scoring

We compute class-discriminative importance by combining curvature-scaled temporal correlation and spatial alignment, incorporating insights from hybrid approaches that fuse GradCAM with Layer-wise Relevance Propagation for enhanced CNN interpretability:

$$I_{\text{time}} = |G_{\text{time}} \cdot A_{\text{time}}| \quad (27)$$
$$I_{\text{space}} = \| \mathcal{G}_{\text{space}} \times_3 \mathcal{A}_{\text{space}} \|_{2,(3)} \quad (28)$$
$$\text{HypGradCAM} = \alpha I_{\text{time}} + \beta I_{\text{space}} \quad (29)$$

The temporal component captures global hierarchical context in the hyperbolic embedding, where the timelike coordinate encodes the scope and generality level of representations. The spatial term uses L2 norms to measure alignment, emphasizing features that are strongly activated and relevant to the class. This combination provides a balanced view: the temporal score grounds the explanation in the manifold's global structure and semantic hierarchy, while the spatial score highlights local discriminative patterns.

The weights $(\alpha, \beta)$ are adjusted by layer depth to balance global contextual information (temporal) with local discriminative information (spatial):

$$(\alpha, \beta) = \begin{cases} (0.05, 1.0) & \text{shallow layers} \\ (0.10, 1.0) & \text{intermediate layers} \\ (0.15, 0.9) & \text{deep layers} \end{cases} \quad (30)$$

In shallow layers, spatial components dominate as they capture local edge and texture patterns. In deeper layers, temporal components become more important as they encode the global semantic context and hierarchical relationships that determine class membership, leveraging the exponential capacity of hyperbolic space to represent multi-scale contextual information.

#### 4.2.3. Sparsity-Aware Emphasis

To maintain visual clarity when sparse activation constraints are imposed, we enhance the spatial importance map with a modulation that compensates for reduced activation spread, inspired by sparsity-aware techniques in recent works like NeurRev [11]:

$$I_{\text{sparse}}^{\text{spatial}} = I^{\text{spatial}} \cdot (1 + 0.2(1 - \rho)) \quad (31)$$

Here, $\rho$ is the sparsity ratio (e.g., fraction of activations kept). Intuitively, as sparsity increases (lower $\rho$), fewer neurons fire, which could make heatmaps too faint. This modulation amplifies the remaining signals, like turning up the volume on key notes in a sparse melody, ensuring important features stand out.

By integrating Hyperbolic GradCAM with our sparsity mechanisms, we visualize how disentangled features emerge in the hyperbolic representation space and assess their contribution to model decisions. This enhanced framework provides more precise, geometry-aware explanations, advancing interpretability in non-Euclidean deep learning.

Intuitively, it's like translating the model's "thought process" from a curved, hierarchical world into flat, understandable visuals, making it easier to see why the model focuses on certain parts of an image.

## 5. Results

In this section, we comprehensively evaluate the impact of sparse activation mechanisms on hyperbolic neural networks. Our analysis proceeds along two main dimensions: (i) quantitative performance, where we measure top-1 classification accuracy across different architectural variants which is trained on the hyperparameters same as in [2], and (ii) interpretability, where we assess model behavior using Hyperbolic GradCAM.

Due to computational limitations, our experiments primarily utilize the ResNet-18 backbone and are assessed on the CIFAR-10 and CIFAR-100 benchmark datasets. We explore Euclidean, fully hyperbolic (Lorentzian), and hybrid architectures, incorporating sparsity via L1 regularization or Top-K activation masking. These evaluations are designed to elucidate not only the performance trade-offs associated with sparsity in hyperbolic networks, but also its impact on the interpretability and structure of the learned representations.

### 5.1. Quantitative Performance Evaluation on CIFAR-10 and CIFAR-100

We evaluate the performance of Euclidean, Lorentzian (fully hyperbolic), and hybrid architectures with and without sparsity mechanisms on CIFAR-10 and CIFAR-100 datasets. Table 1 reports Top-1 accuracy (%) for each variant. Sparsity is introduced using L1 regularization or Top-K masking, and the hybrid model follows the configuration described in [2] where blocks with high hyperbolicity (e.g., 1 and 3) are replaced with Lorentz blocks while others remain Euclidean.

These results demonstrate that hyperbolic geometry facilitates compact, expressive representations, with sparsity introducing negligible performance degradation while providing greater interpretability as shown in subsection 5.2.

### 5.2. Hyperbolic GradCAM analysis

To assess the qualitative interpretability benefits of hyperbolic models, we visualize the GradCAM heatmaps generated from Euclidean and fully hyperbolic CNNs. Figure 2 shows comparisons on the same input image. We observe that while the Euclidean GradCAM tends to produce broader, often diffused attention regions that may highlight irrelevant background areas, the Hyperbolic GradCAM yields sharper, spatially localized, and semantically focused activations, concentrating more effectively on the discriminative regions (e.g., the contours and head of the

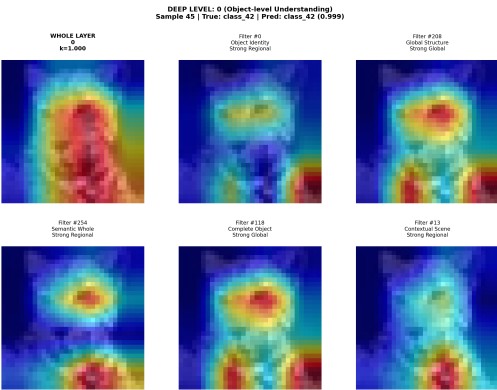

Figure 1. The Top Filters in the deeper layer of Lorentz CNNs where the first image is the whole object

frog).

We hypothesize that this difference stems from the hyperbolic model's intrinsic capacity to encode hierarchical relations. Instead of merely identifying low-level discriminative patterns, the hyperbolic geometry allows the network to learn global structural cues those that define what makes an object a "frog" in a taxonomic or conceptual sense, beyond superficial texture or contrast differences. This aligns with the theory that hyperbolic spaces are better suited to represent hierarchical or tree-like data structures [7, 20].

### 5.3. Filters level analysis of layers

To understand what individual filters are learning and how this is aggregated into the whole, we conduct a systematic analysis of filter–layer relationships at the layer level, as these relationships constitute fundamental evidence of object understanding within neural architectures. Individual convolutional filters function as part detectors owing to their localized receptive fields, which exhibit selective responses to specific visual patterns and structural components. Conversely, aggregated layer activations elucidate whole-object comprehension through the principled integration of distributed part responses across spatial feature maps.

Our analytical framework decomposes the Lorentz manifold representation for GradCAM into temporal and spatial components, formulating a weighted combination as $CAM = \alpha \cdot I_{\text{time}} + \beta \cdot I_{\text{space}}$. The temporal components encapsulate global contextual dependencies leveraging the manifold's timelike dimension, whereas spatial components encode local geometric part relationships through spacelike dimensions.

Filter significance is rigorously quantified through gradient-activation products, importance $= \text{mean}(|\nabla f \times \text{activation}_f|)$, which simultaneously measure classification sensitivity and feature presence to evaluate each filter's contribution to global understanding. Through comprehensive visualization demonstrating how individual filters de-

| Domain | Variant | CIFAR-10 | CIFAR-100 |
|--------|---------|----------|-----------|
| Euclidean | Baseline (ResNet-18) | 95.14±0.12 | 77.72±0.15 |
| | + L1 Sparse (all layers) | 95.21±0.11 | 77.75±0.12 |
| | + Top-K Sparse ($\rho = 0.1$) | 95.19±0.15 | 77.53±0.12 |
| | + Top-K Sparse ($\rho = 0.01$) | **95.49±0.11** | **77.81±0.11** |
| Lorentz | Baseline (Hyp-ResNet18) | 95.14±0.08 | 78.07±0.17 |
| | + L1 Sparse | 95.11±0.13 | 77.52±0.21 |
| | + Top-K Sparse ($\rho = 0.1$) | 95.14±0.09 | **78.09±0.11** |
| | + Top-K Sparse ($\rho = 0.01$) | **95.14±0.13** | 78.02±0.19 |
| Hybrid | Baseline (Hybrid ResNet) | 95.16±0.11 | **78.56±0.24** |
| | + L1 Sparse | 95.23±0.18 | 78.01±0.07 |
| | + Top-K Sparse ($\rho = 0.1$) | **95.31±0.02** | 77.81±0.06 |
| | + Top-K Sparse ($\rho = 0.01$) | 95.25±0.06 | 78.00±0.12 |

Table 1. Top-1 accuracy (%) on CIFAR-10 and CIFAR-100 across Euclidean, Lorentzian, and Hybrid variants with different sparsity mechanisms. Top-K sparsity at $\rho = 0.01$ achieves the best performance in Euclidean settings, while hybrid and Lorentzian models show strong results on CIFAR-100.

tect constituent object parts while layer-wise aggregations capture holistic semantic representations, we show how filter wise part information is aggregated into the layer output. We also observe a hierarchy emerge when examining activations across layers (refer supp.), though these are not interpretable enough to make conclusive comments on the part-whole hierarchy across layers.

### 5.4. Analysis of activation sparsity in Hyperbolic CNN

We investigate whether sparse activations encourage the network to focus on the most critical, high-salience features, thereby enhancing interpretability without compromising performance.

This line of inquiry is grounded in the hypothesis that activation sparsity can act as a form of structural inductive bias, promoting disentanglement in the latent space and improving the selectivity of GradCAM attributions. In doing so, we aim to bridge architectural expressiveness (via hyperbolic geometry) with functional parsimony (via sparsity), both of which are known to contribute to interpretable representations in biological systems [8].

Figure 2 demonstrates the qualitative effects of applying sparsity to hyperbolic CNNs via L1 and Top-$k$ activation constraints. Across all configurations, we observe a consistent sharpening of GradCAM heatmaps as sparsity increases. Specifically:

**L1 Sparse Hyperbolic GradCAM** shows moderately focused attention with denoised activations that remain semantically relevant and follow object contours.
**Top-$k$ Sparse** variants highlight salient object regions more aggressively, producing concentrated and interpretable maps.

**Harder Top-$k$** (with lower $\rho$) further localizes attention to core features, although occasionally at the cost of contextual cues.

In hyperbolic space, points near the origin encode more global, high-level concepts, while those farther from the origin correspond to fine-grained, leaf-level details. L1 sparsity, by penalizing all activations uniformly, tends to shrink embeddings toward the origin. This implicitly favors the preservation of global, category-level information, but at the cost of pruning away peripheral, detail-rich features. Consequently, GradCAM maps under L1 sparsity appear smoother and less sharply localized, reflecting an emphasis on broader semantic coverage rather than precise spatial delineation. In contrast, Top-$k$ sparsity explicitly retains activations that lie farther from the origin, thereby preserving the periphery of the hyperbolic space where the most discriminative and specific features reside. This selection bias maintains the leaf-level distinctions in the hierarchy, producing activation patterns that are both more selective and more spatially precise. This can actually be also seen from the accuracies of the model which are higher for Top-$k$ models 1. As seen in GradCAM, this results in sharply concentrated heatmaps that tightly follow object boundaries, enhancing fine-grained interpretability. Harder Top-$k$ thresholds further accentuate this effect, focusing attention almost exclusively on the most discriminative structures, though with a potential reduction in contextual awareness.

### 5.5. Quantitative metrics for GradCAM analysis

To better understand the interpretability benefits of sparse activation mechanisms, we evaluate GradCAM-based vi-

Table 2. GradCAM evaluation metrics for layer 15 across standard and sparse hyperbolic networks trained on CIFAR-100. Bold values indicate best performance per metric (excluding complexity, where lower is better).

| Model | Robustness ↑ | Faithfulness ↑ | Localization ↑ | Complexity ↓ | Interpretability ↑ |
|-------|-------------|----------------|----------------|--------------|-------------------|
| Standard | $0.556 \pm 0.173$ | $0.148 \pm 0.096$ | $0.062 \pm 0.046$ | $-16.952 \pm 5.278$ | $0.682 \pm 0.025$ |
| L1 Sparse | $\mathbf{0.702 \pm 0.134}$ | $\mathbf{0.233 \pm 0.068}$ | $0.063 \pm 0.032$ | $\mathbf{-17.807 \pm 2.113}$ | $\mathbf{0.699 \pm 0.023}$ |
| Top-0.1% Sparse | $0.699 \pm 0.154$ | $0.140 \pm 0.102$ | $\mathbf{0.066 \pm 0.037}$ | $-16.262 \pm 5.344$ | $0.664 \pm 0.047$ |
| Top-0.01% Sparse | $0.694 \pm 0.158$ | $0.140 \pm 0.102$ | $\mathbf{0.066 \pm 0.037}$ | $-16.262 \pm 5.344$ | $0.664 \pm 0.047$ |

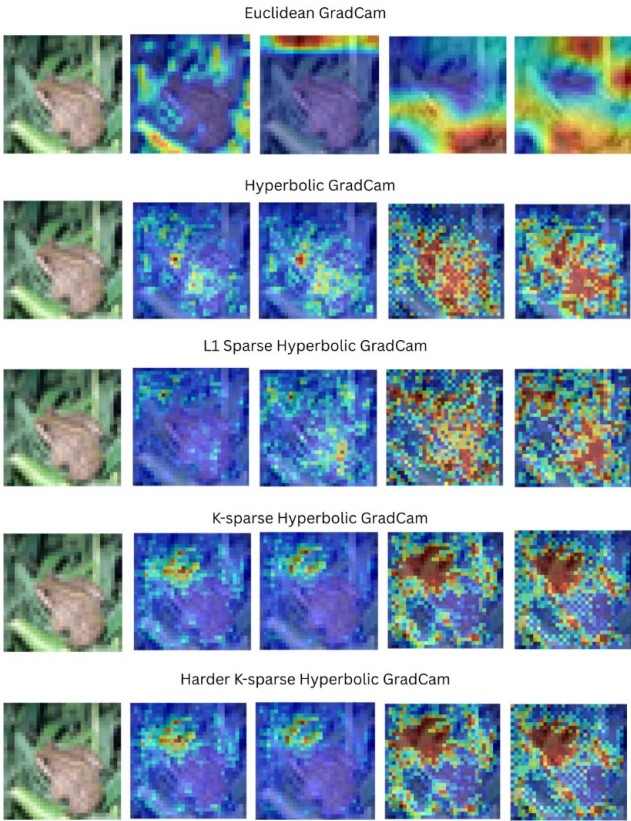

Figure 2. GradCAM visualizations for hyperbolic CNNs with different activation sparsity mechanisms. From top to bottom: L1 sparse, Top-$k$ sparse ($\rho = 0.1$), and harder Top-$k$ sparse ($\rho = 0.01$). Each row shows the original image followed by activation maps from successive layers.

sual explanations using five key metrics: **Robustness**, **Faithfulness**, **Localization**, **Complexity**, and **Interpretability**. *Robustness* measures the stability of the saliency maps under perturbations, where higher values imply more consistent explanations. *Faithfulness* quantifies how well the saliency map aligns with the model's true decision-making process (e.g., via input occlusion). *Localization* evaluates the sharpness and spatial concentration of salient regions, indicating how focused the explanations are. *Complexity*, in contrast, is minimized; more negative values denote simpler and less noisy saliency maps. Finally, *Interpretability* is an aggregate score indicating how comprehensible the explanations are to humans, combining fidelity and sparsity-based heuristics.

From Table 2, it is evident that sparse variants, especially the **L1 Sparse** model, outperform the standard hyperbolic network across most metrics. It achieves the highest **Robustness**, **Faithfulness**, and **Interpretability**, while also having the lowest (i.e., best) **Complexity**. Interestingly, both **Top-0.1%** and **Top-0.01%** sparsity levels exhibit superior **Localization** scores compared to the baseline, suggesting sharper and more spatially focused attention maps. We have also provided results for all layers across training regimes for hyperbolic models in the supplementary section.

These results provide compelling evidence that sparse hyperbolic networks not only preserve but often enhance interpretability across multiple axes. This underscores a strong case for further investigating sparse activation mechanisms as principled methods for improving model transparency and alignment with cognitively relevant priors.

# 6. Conclusion and discussion

We introduce Hyperbolic GradCAM, an interpretability framework extending gradient-based visual explanations to hyperbolic CNNs by incorporating Lorentzian geometry and separating spatiotemporal components. This enables principled visualizations of hyperbolic models.

In parallel, we study sparse hyperbolic CNNs using $L_1$ regularization and Top-$K$ activation masking. These models match the performance of Euclidean and hyperbolic baselines while producing sharper, more meaningful attention maps under Hyperbolic GradCAM.

Our results show that combining hyperbolic representations with sparse activations yields more expressive and interpretable models. Future work will investigate how sparsity promotes disentanglement in hyperbolic feature spaces, paving the way for structured, semantically aligned explanations in non-Euclidean deep learning.

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
