# Sparse Hyperbolic Convolutional Networks with Enhanced Object Localization via GradCAM Analysis

## Supplementary Material

## 1. GradCAM examples for Hyperbolic

Here we provide 2 more example classes across all 18 layers of the ResNet-18 network.

## 2. GradCAM evaluation metrics for all layers for all hyperbolic model

Table 1. Grad-CAM Evaluation Metrics Across Training Methods

| Layer | Robustness | | | | Faithfulness | | | | Localisation | | | | Complexity | | | | Interpretability | | | |
|---|---|---|---|---|---|---|---|---|---|---|---|---|---|---|---|---|---|---|---|---|
| | Top-0.01% | Top-0.1% | L1 | Normal | Top-0.01% | Top-0.1% | L1 | Normal | Top-0.01% | Top-0.1% | L1 | Normal | Top-0.01% | Top-0.1% | L1 | Normal | Top-0.01% | Top-0.1% | L1 | Normal |
| 1 | 0.502 | 0.499 | 0.449 | 0.465 | 0.147 | 0.147 | 0.162 | 0.149 | 0.030 | 0.029 | 0.017 | 0.017 | -12.749 | -12.749 | -13.729 | -12.100 | 0.922 | 0.922 | 0.920 | 0.921 |
| 2 | 0.422 | 0.428 | 0.340 | 0.407 | 0.186 | 0.186 | 0.161 | 0.155 | 0.026 | 0.027 | 0.014 | 0.017 | -12.871 | -12.871 | -12.516 | -14.106 | 0.911 | 0.911 | 0.904 | 0.918 |
| 3 | 0.597 | 0.596 | 0.510 | 0.558 | 0.181 | 0.181 | 0.177 | 0.184 | 0.025 | 0.026 | 0.012 | 0.016 | -11.999 | -11.999 | -11.933 | -12.097 | 0.906 | 0.906 | 0.899 | 0.905 |
| 4 | 0.551 | 0.560 | 0.441 | 0.520 | 0.187 | 0.187 | 0.181 | 0.176 | 0.025 | 0.026 | 0.011 | 0.015 | -10.719 | -10.719 | -9.125 | -11.836 | 0.918 | 0.918 | 0.903 | 0.918 |
| 5 | 0.705 | 0.703 | 0.657 | 0.678 | 0.194 | 0.194 | 0.177 | 0.179 | 0.035 | 0.034 | 0.016 | 0.019 | -13.186 | -13.186 | -10.088 | -12.994 | 0.897 | 0.897 | 0.872 | 0.897 |
| 6 | 0.729 | 0.742 | 0.629 | 0.654 | 0.221 | 0.221 | 0.171 | 0.189 | 0.036 | 0.037 | 0.014 | 0.020 | -7.825 | -7.825 | -6.663 | -9.490 | 0.892 | 0.892 | 0.878 | 0.901 |
| 7 | 0.764 | 0.774 | 0.698 | 0.704 | 0.250 | 0.250 | 0.207 | 0.209 | 0.034 | 0.035 | 0.013 | 0.018 | -8.174 | -8.174 | -6.211 | -8.307 | 0.884 | 0.884 | 0.869 | 0.892 |
| 8 | 0.688 | 0.695 | 0.626 | 0.652 | 0.260 | 0.260 | 0.187 | 0.208 | 0.038 | 0.039 | 0.014 | 0.020 | -8.681 | -8.681 | -6.612 | -8.807 | 0.892 | 0.892 | 0.871 | 0.892 |
| 9 | 0.767 | 0.797 | 0.708 | 0.702 | 0.260 | 0.260 | 0.196 | 0.201 | 0.040 | 0.042 | 0.014 | 0.020 | -8.952 | -8.952 | -6.290 | -8.562 | 0.880 | 0.880 | 0.854 | 0.876 |
| 10 | 0.832 | 0.828 | 0.755 | 0.752 | 0.263 | 0.263 | 0.208 | 0.214 | 0.041 | 0.043 | 0.017 | 0.022 | -7.413 | -7.413 | -6.847 | -7.702 | 0.821 | 0.821 | 0.812 | 0.828 |
| 11 | 0.871 | 0.861 | 0.836 | 0.793 | 0.254 | 0.254 | 0.215 | 0.228 | 0.045 | 0.046 | 0.018 | 0.024 | -8.081 | -8.081 | -7.436 | -8.601 | 0.819 | 0.819 | 0.814 | 0.828 |
| 12 | 0.871 | 0.877 | 0.883 | 0.852 | 0.254 | 0.254 | 0.235 | 0.211 | 0.059 | 0.060 | 0.031 | 0.041 | -10.548 | -10.548 | -9.254 | -11.315 | 0.830 | 0.830 | 0.822 | 0.827 |
| 13 | 0.867 | 0.857 | 0.892 | 0.831 | 0.229 | 0.229 | 0.236 | 0.229 | 0.049 | 0.051 | 0.032 | 0.039 | -11.205 | -11.205 | -10.861 | -11.882 | 0.839 | 0.839 | 0.837 | 0.845 |
| 14 | 0.900 | 0.888 | 0.903 | 0.811 | 0.215 | 0.215 | 0.224 | 0.195 | 0.040 | 0.041 | 0.029 | 0.022 | -15.305 | -15.305 | -15.470 | -14.678 | 0.719 | 0.719 | 0.714 | 0.726 |
| 15 | 0.694 | 0.699 | 0.702 | 0.556 | 0.140 | 0.140 | 0.233 | 0.148 | 0.066 | 0.066 | 0.048 | 0.061 | -16.262 | -16.262 | -17.807 | -16.952 | 0.664 | 0.664 | 0.699 | 0.682 |
| 16 | 0.820 | 0.829 | 0.769 | 0.807 | 0.187 | 0.187 | 0.210 | 0.218 | 0.034 | 0.036 | 0.018 | 0.022 | -13.994 | -13.994 | -13.900 | -15.688 | 0.689 | 0.689 | 0.696 | 0.695 |
| 17 | 0.694 | 0.699 | 0.702 | 0.556 | 0.140 | 0.140 | 0.233 | 0.148 | 0.080 | 0.083 | 0.061 | 0.072 | -16.262 | -16.262 | -17.807 | -16.952 | 0.664 | 0.664 | 0.699 | 0.682 |

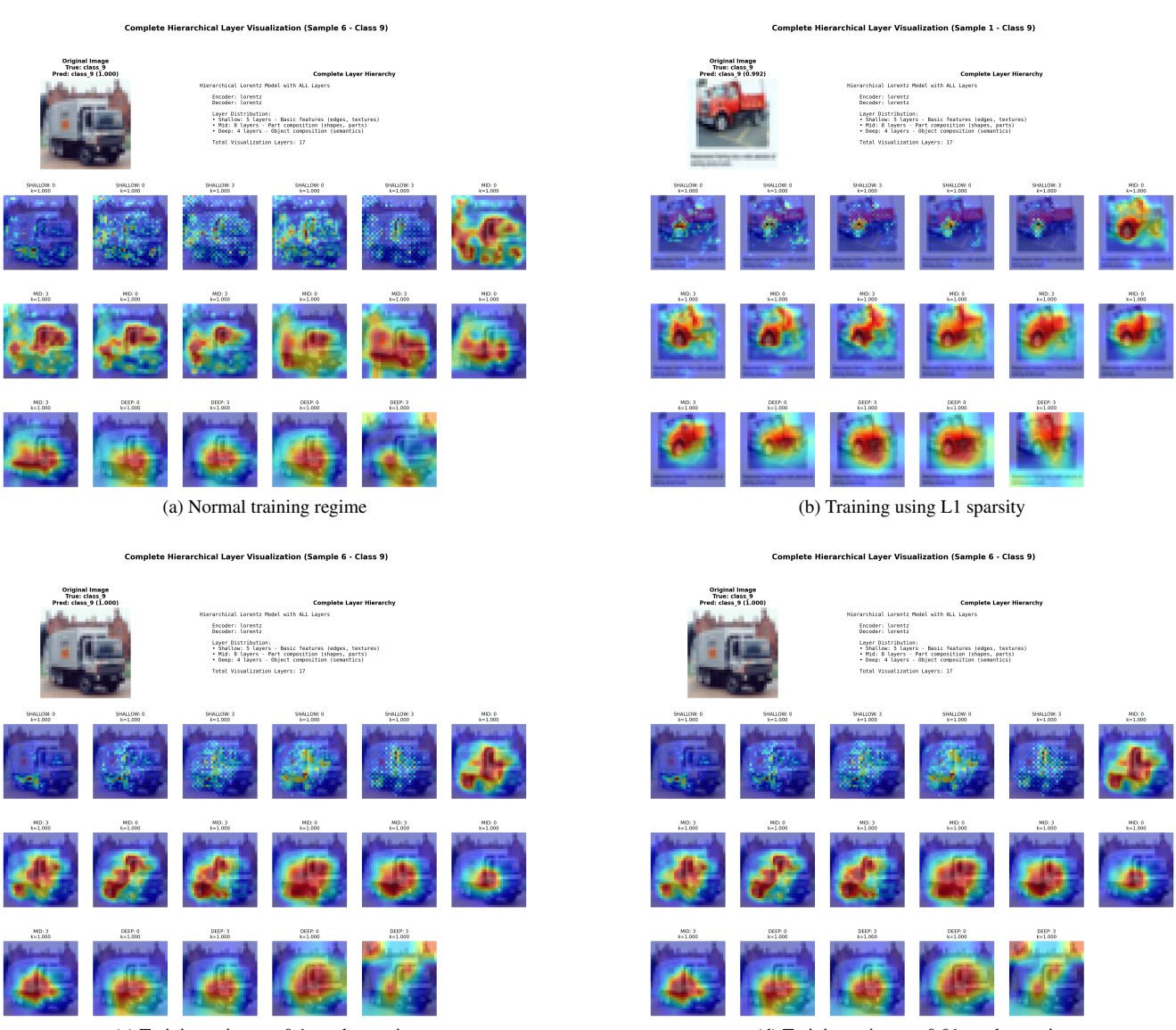

(a) Normal training regime

(b) Training using L1 sparsity

(c) Training using top 0.1 top-k sparsity

(d) Training using top 0.01 top-k sparsity

Figure 1. GradCAM analysis of all 18 layers of resnet under different training regimes

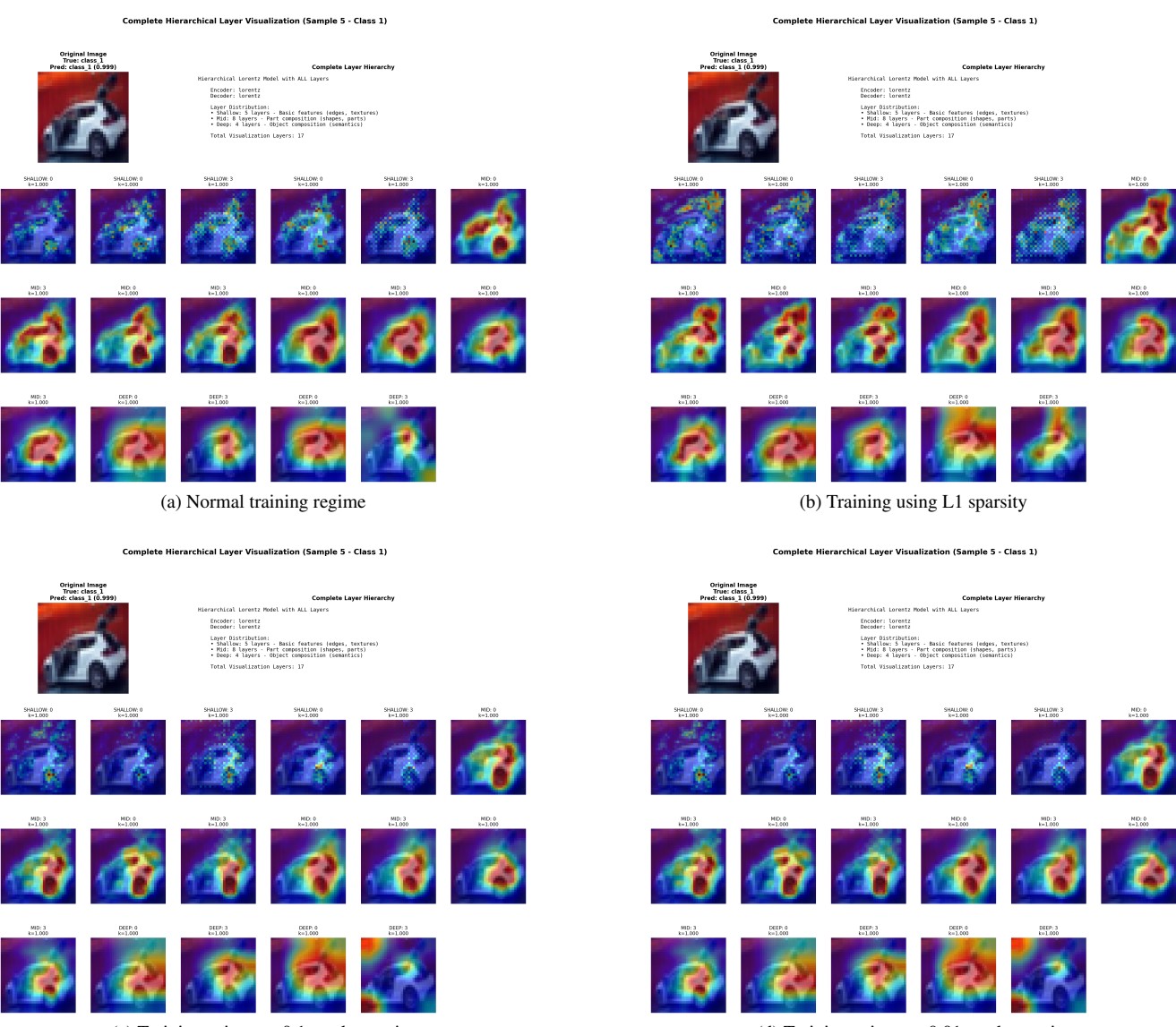

Figure 2. GradCAM analysis of all 18 layers of resnet under different training regimes

**DEEP LEVEL: 0 (Object-level Understanding)**
**Sample 20 | True: class_21 | Pred: class_21 (0.999)**

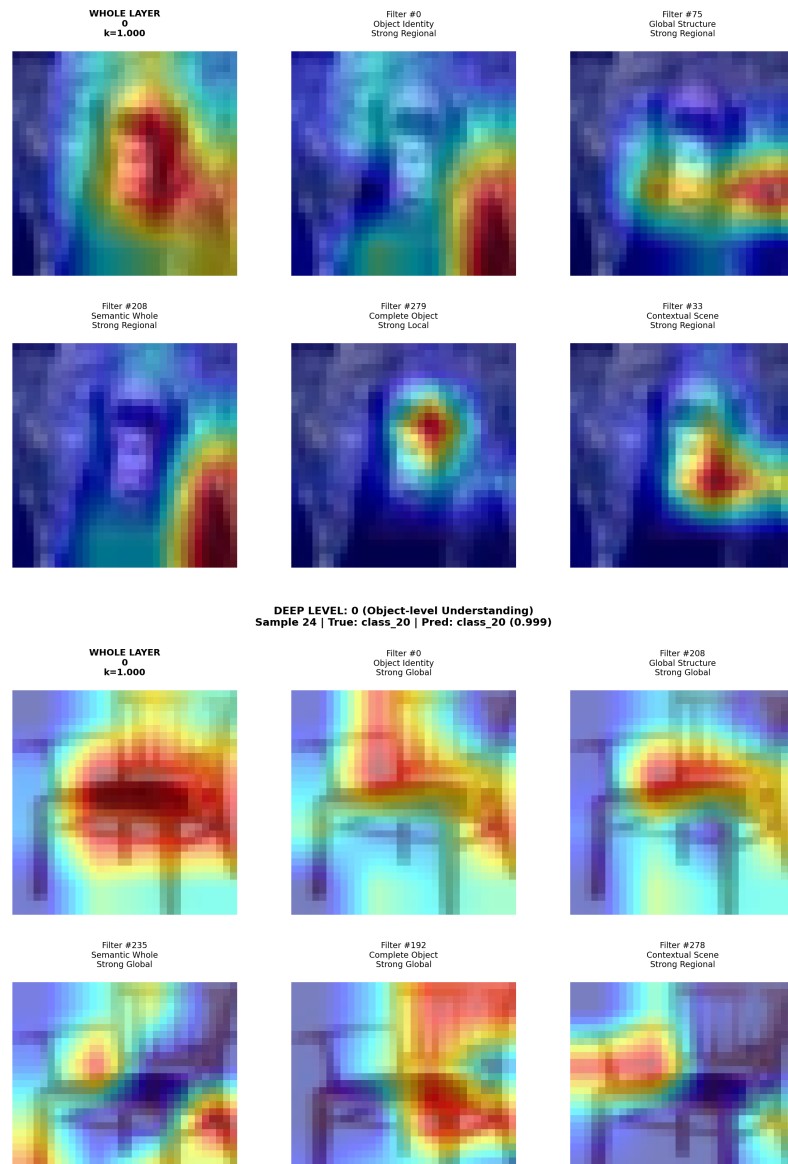

**DEEP LEVEL: 0 (Object-level Understanding)**
**Sample 24 | True: class_20 | Pred: class_20 (0.999)**

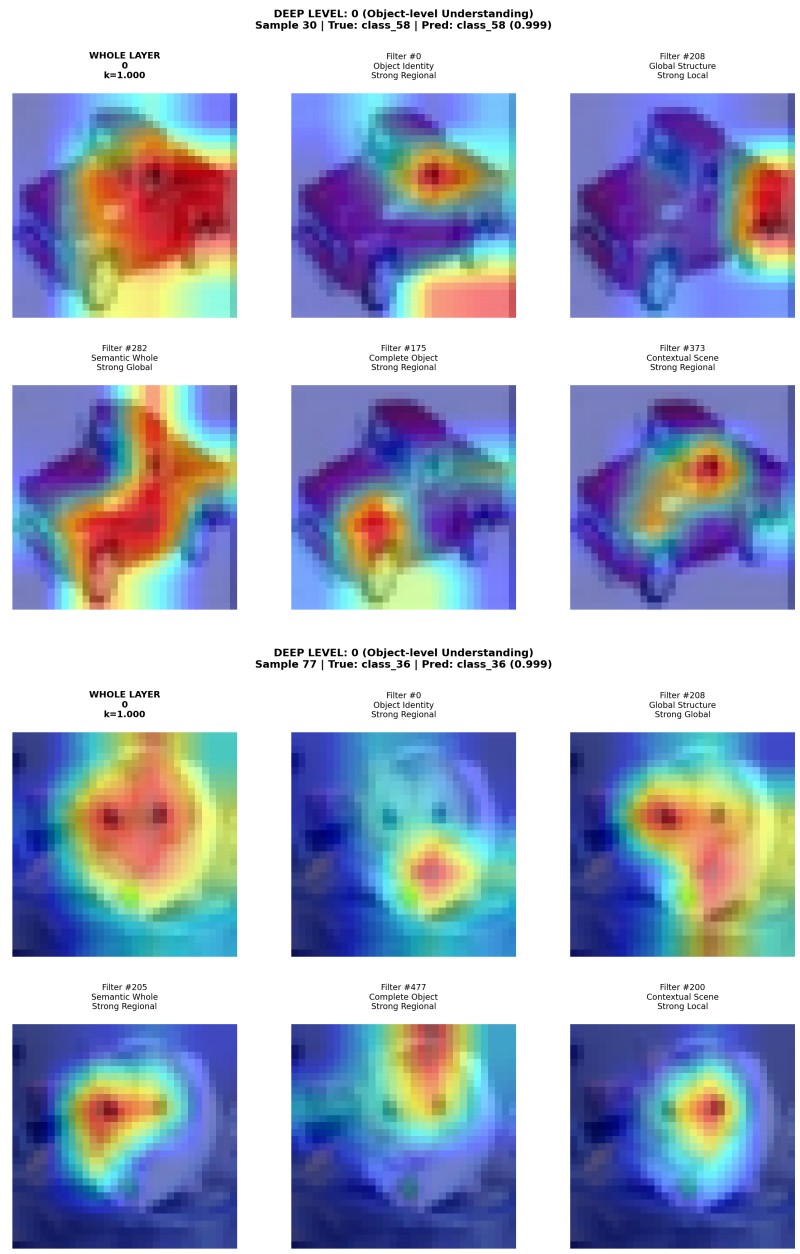

Figure 4. Comprehensive Heatmap of the layer and Top-5 Filters Ranked by Individual GradCAM Contributions

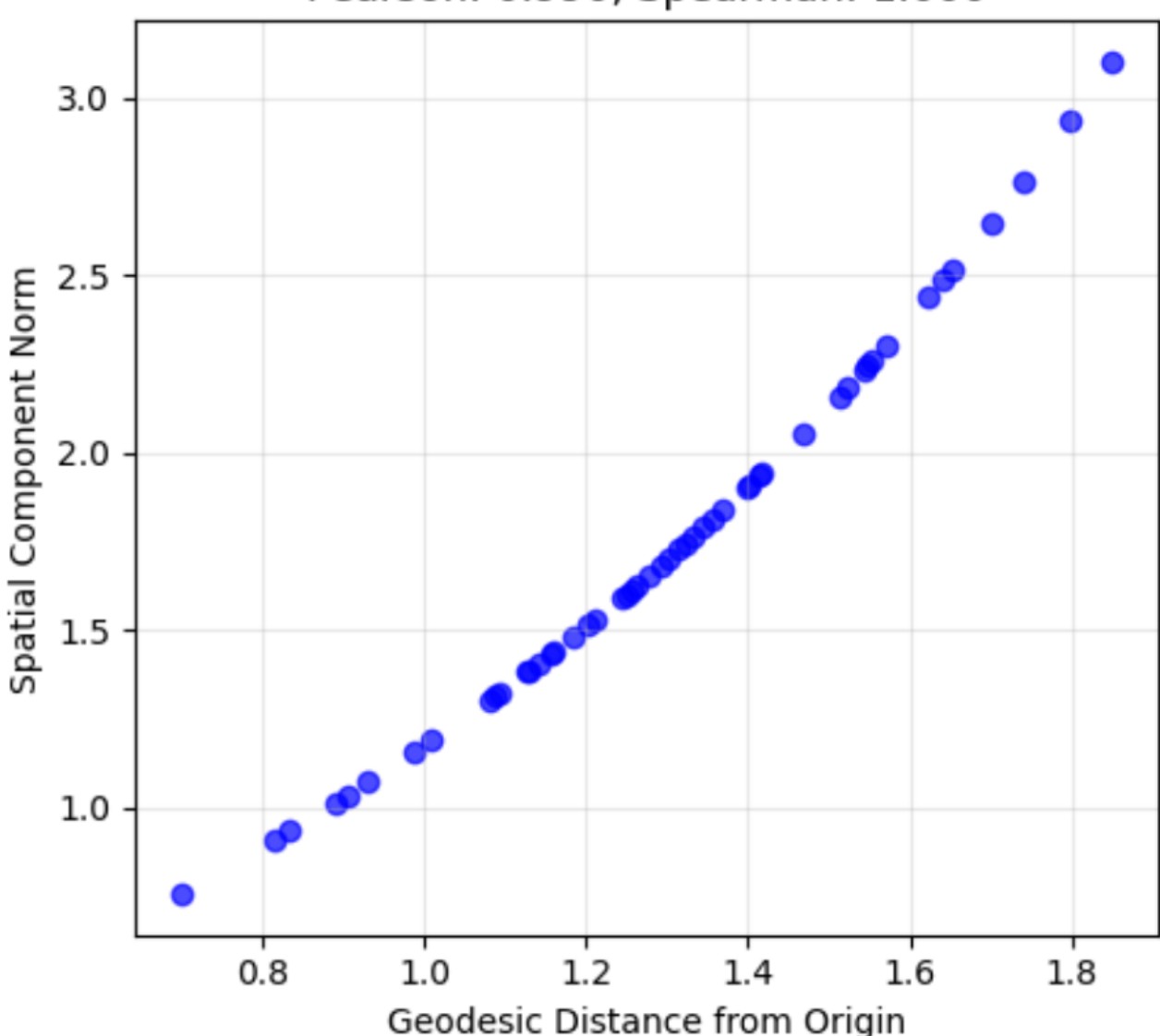

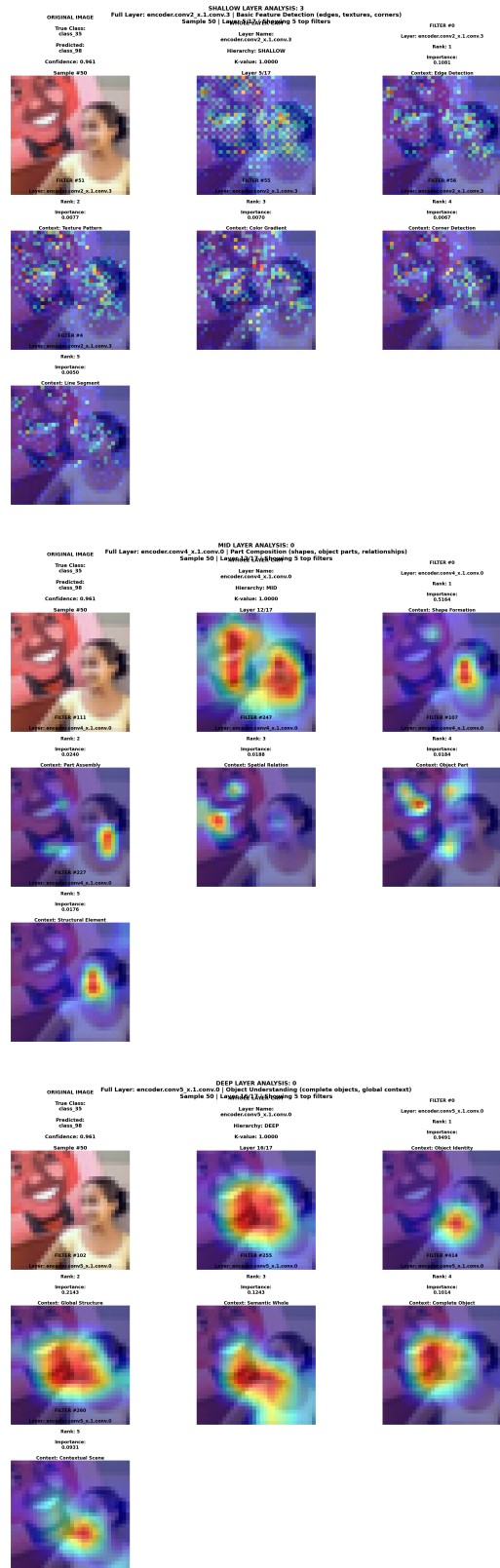

Figure 5. GradCAM images from shallow, mid and deep layers along with the top filters