# OpenReview forum: "Sparse Hyperbolic Convolutional Networks with Enhanced Object Localization via GradCAM Analysis"
_thecvf.com/ICCV/2025/Workshop/BEW — BEW 2025 Poster_

### Official Review · Reviewer_sU5s · 2025-06-30
**Hyperbolic convolutional models with sparse regularization**

**Rating:** 4
**Confidence:** 3

**Review:**

This paper extends Hyperbolic convolutional models to sparse models employing sparse regularization. The methodology looks fine, as the contribution was to add sparse penalties to the original models. The results supported the proposal. I am not an expert in this area. In the figure 1 (as well as Figure 2), what are the four heatmaps next to the original images? It looks like that the heatmaps are indicating more about the objects using the sparse models compared to the Euclidean model. What was the intuition or theoretical reasoning why sparse regularization helped that? Why was the Hybrid baseline method (without sparsity) doing the best across all the domains on CIFAR-100 in Table 1? Why was Lorentz baseline was doing so badly (=8.00)?

---

### Official Review · Reviewer_ekGR · 2025-07-06
**This paper introduces Sparse Hyperbolic CNNs with a novel Hyperbolic GradCAM for enhanced interpretability. It shows hyperbolic CNNs produce more interpretable activations, but sparsity's benefits are unclear, and the visual evidence is limited.**

**Rating:** 2
**Confidence:** 3

**Review:**

## Strengths

- The paper is sufficiently well-written and easy to follow. While some sections could benefit from more depth, the overall presentation is clear.
- The proposed Hyperbolic GradCam is novel and it seems a promising tool for interpretability in hyperbolic neural networks. The visual analyses presented in the paper show that fully hyperbolic CNNs produce more interpretable activations, focusing on the object of interest in the image, compared to their Euclidean counterpart.

## Weaknesses

- Applying L1 regularization in the tangent space rather than directly in hyperbolic space raises questions about its usefulness. While L1 regularization is known to work well in Euclidean settings, its impact in hyperbolic space remains unclear. The results in Table 1 don't show clear benefits from sparsity in either Euclidean or hyperbolic models, as sparse variants merely match dense performance. To strengthen these findings, multiple runs with different seeds and standard error calculations would be necessary to confirm any subtle improvements.

- The authors assert that sparsity improves interpretability, but this claim isn't convincingly demonstrated. Figure 2 compares different sparsity-inducing methods but doesn't show sparse vs. dense comparisons, making it hard to isolate sparsity's effect. If sparsity doesn't enhance interpretability or performance, its practical value for HCNNs becomes questionable. Providing side-by-side comparisons of sparse and dense models would help validate the claim.

- The visual analyses rely heavily on a single frog image and a boat image, which feels unsufficient and potentially cherry-picked. Including a broader range of examples would make the findings more robust and generalizable.

- The paper lacks references to recent work (2024–2025) on hyperbolic neural networks, which is a missed opportunity to contextualize the contributions. Updating the related work section with recent advancements would be appreciated.

- The authors suggest that sparse hyperbolic networks maintain competitive accuracy while using fewer computational resources, making them attractive for resource-constrained applications. However, it lacks a comparisons of computational metrics like GFLOPS, memory usage, or number of parameters between sparse and dense variants to back up the claim.

## Minor Points

- The sentence "Studies have demonstrated that spatial representations in the hippocampus are structured within hyperbolic spaces to optimize efficiency [17]" feels out of place in the abstract. It doesn't clearly connect to the paper's main focus, and removing it would allow space for more relevant context or contributions to be highlighted.
- A citation error (i.e., a question mark "?") appears on line 143 and should be corrected.
- The citation for [2] should be updated to reflect its publication at ICLR 2024.
- The value "8.00" in Table 1 (Lorentz-Baseline-CIFAR-100) seems like a typo and should be verified.

---

### Official Review · Reviewer_koyh · 2025-07-07
**Technical implementation seems solid, but impact is doubtful, thus I recommend weak reject.**

**Rating:** 2
**Confidence:** 5

**Review:**

**Summary**

This paper introduces Hyperbolic GradCAM, a post-hoc interpretability framework adapted to Lorentz geometry. The authors incorporate sparsity through L1 regularization and Top-k masking into hyperbolic CNNs, aiming to improve both computational efficiency and interpretability. They also propose a curvature-aware GradCAM extension that decomposes activations and gradients into temporal and spatial components, enabling saliency visualization that respects hyperbolic geometry. Experiments show that sparse hyperbolic networks maintain competitive accuracy and yield sharper, more sparse heatmaps compared to Euclidean baselines.

**Justification**

Although some choices (soft thresholding, decomposition of temporal-spatial features, etc) need to be more elaborated, the technical implementation in the Method section seems solid. However, the interpretability claims in hyperbolic space remain limited in scope. The authors hypothesize that hyperbolic geometry enables the model to capture conceptual structure, allowing it to identify semantically meaningful global features. This is a reasonable and theoretically grounded assumption, but it is not rigorously tested in the experiments. One would expect the GradCAM outputs to reveal heatmaps that reflect hierarchical structure, such as part-whole relationship discovery within the object. However, this aspect is not thoroughly addressed, which narrows the interpretive contribution of the proposed GradCAM extension for hyperbolic CNNs.

---

### Decision · Program_Chairs · 2025-07-09

**Decision:**

Accept (Poster)

**Comment:**

The claims about interpretability in the hyperbolic domain are limited and we strongly advise the authors to enhance this part during the camera ready preparations.  Moreover, the literature should be updated.